# Enhanced Anxiety and Olfactory Microglial Activation in Early-Stage Familial Alzheimer’s Disease Mouse Model

**DOI:** 10.3390/biology11060938

**Published:** 2022-06-20

**Authors:** Keerthana Chithanathan, Fang-Ling Xuan, Miriam Ann Hickey, Li Tian

**Affiliations:** 1Department of Physiology, Faculty of Medicine, Institute of Biomedicine and Translational Medicine, University of Tartu, Ravila 14b, 50411 Tartu, Estonia; keerthana.chithanathan@ut.ee (K.C.); fangling.xuan@ut.ee (F.-L.X.); 2Department of Pharmacology, Faculty of Medicine, Institute of Biomedicine and Translational Medicine, University of Tartu, Ravila 19, 50411 Tartu, Estonia; miriam.ann.hickey@ut.ee

**Keywords:** Alzheimer’s disease, olfactory bulb, anxiety, microglia, synaptic pruning, neuroinflammation

## Abstract

**Simple Summary:**

We observed that compared to wildtype (WT) littermates, 5 × FAD mice showed enhanced anxiety with concomitant increased pro-inflammatory cytokines in the olfactory bulb (OB) but not the frontal cortex (FC) at as early as 2 months old (mo). More prominent microglial activation and morphological changes were also found in the OB of 2 mo 5 × FAD mice. In the FC, pro-inflammatory cytokines were upregulated at the later 5~6 mo stage. Furthermore, myeloid cell number and microglial phagocytosis of presynaptic vesicular glutamate transporter-2 were increased in the OB but not the FC of 5~6 mo 5 × FAD mice. Our findings demonstrated that microglia-mediated neuroinflammation in the OB can be an early-stage biomarker for neuropsychiatric behavior in AD.

**Abstract:**

Anxiety is a known comorbidity and risk factor for conversion to neuroinflammation-mediated dementia in patients with Alzheimer’s disease (AD). Here, we investigated if anxiety occurred as an early endophenotype of mutant familial AD (5 × FAD) male mice and the underlying neuroinflammatory mechanisms. We observed that compared to wildtype (WT) littermates, 5 × FAD mice showed enhanced anxiety at as early as 2 months old (mo). Interestingly, these 5 × FAD male mice had concomitantly increased mRNA levels of pro-inflammatory cytokines such as interleukin 1 beta (*Il1b*) and tumor necrosis factor (*Tnf*) in the olfactory bulb (OB) but not the frontal cortex (FC). Increased expression of *Tnf* in the OB was significantly correlated with the anxious behavior in the FAD but not WT mice. Furthermore, we found more prominent microglial activation and morphological changes in the OB of 2 mo 5 × FAD mice, while only microglial ramification was seen in the FC. To understand if neuroinflammatory changes in the FC could occur at a later stage, we studied 5~6 mo male mice and found that *Il1b*, interleukin 18 (*Il18*), and *Tnf* were upregulated in the FC at this older age. Furthermore, we observed that numbers of microglia and macrophage as well as microglial synaptic pruning, as indicated by phagocytosis of presynaptic component of vesicular glutamate transporter-2, were increased in the OB but not the FC of 5~6 mo 5 × FAD mice. Our findings demonstrated the OB as a more sensitive brain region than the cerebral cortex for microglia-mediated neuroinflammation in association with anxiety in FAD mice and supported the notion that the OB can be an early-stage biomarker in AD.

## 1. Introduction

Alzheimer’s disease (AD) is an increasingly prevalent disease and a leading cause of dementia in the aged population**,** with heavy healthcare, societal, and economical burdens worldwide. It is primarily expressed as neurocognitive decline and is also accompanied by prominent neuropsychiatric symptoms, including anxiety and depression [1]. Anxiety is an early affective symptom in AD patients, and the prevalence of anxiety in AD is about 40% [2]. Anxiety contributes to the progression from prodromal mild cognitive impairment to AD [3], accompanies cognitive impairment at early-stage AD [4,5], and accelerates cognitive decline in AD patients at later stages [6]. An important factor affecting the incidence of anxiety in AD is the age of disease onset, as research has demonstrated the occurrence of psychological symptoms in AD patients with early-onset (i.e., onset before 65 years of age) compared to those with late-onset disease [7].

Notably, olfactory dysfunction is another early symptom commonly observed in AD patients or animal models, preceding the clinical or preclinical manifestation of dementia [8]. Notably, AD patients with anxiety were found to show olfactory impairments, such as worse odor discrimination, which was correlated with the severity of anxiety [9]. In rodents, olfactory deficits caused by damages such as olfactory bulbectomy leads to enhanced anxiety and depressive-like behaviors, as constantly reported by others and us [10,11,12], demonstrating a central behavior-modulatory role of olfactory sensation in rodents. Anxiety and olfactory dysfunction may reciprocally affect each other as brain regions controlling emotional and olfactory processes are closely linked, so anxiety causes changes in smell and taste, and conversely, loss of smell and taste causes distress and anxiety. Hence, olfactory dysfunction is a possible link to anxiety in early-stage AD. However, scientific evidence supporting this notion does not yet exist and warrants more attention of research.

Early-onset AD can be recapitulated in animal models such as 5× familial AD (5 × FAD) mice, which express human *APP* and *PSEN1* transgenes harboring five AD-linked genetic mutations that phenocopy human-like AD pathologies and dementia [13]. In this model, numerous studies have observed the amyloidosis, neuropathy, and gliosis-induced neuroinflammation in brain regions, including the olfactory bulb (OB), as well as the associated cognitive deficits to occur after 4~6 months of age [13,14,15,16,17,18]. In comparison, much less has been studied at the earlier stage of this animal model. Nevertheless, cerebral amyloidosis was reported at as young as 1.5 months of age in this model [14], indicating the potential occurrence of brain and behavioral deficits in young 5 × FAD mice.

Since little is known about young 5 × FAD mice, to fill this knowledge gap and to clarify if behavioral endophenotype is associated with glia-induced neuroinflammation in this model, we studied 2~6-month-old (mo) 5 × FAD male mice and observed an interesting presentation of anxiety behaviors with concomitant neuroinflammatory changes in the OB and cerebral frontal cortex (FC) of young FAD male mice.

## 2. Materials and Methods

### 2.1. Animals

Transgenic 5 × FAD male mice of 2 mo (*n* = 7~8) and 5~6 mo (*n* = 5) and their paired wildtype (WT) littermates (Jackson laboratory, MMRRC Stock No: 34840-JAX) were group-housed under standard conditions with unlimited access to food and water on a 12/12 h light/dark cycle in laboratory animal facility at the Institute of Biomedicine and Translational Medicine, University of Tartu (License No. 175). Tail tips were taken for genotyping, and mice were genotyped using the following primers: *PSN1* forward AATAGAGAACGGCAGGAGCA; *PSN1* reverse GCCATGAGGGCACTAATCAT; WT *APP* forward CTAGGCCACAGAATTGAAAGATCT; WT *APP* reverse GTAGGTGGAAATTCTAGCATCATCC [15].

### 2.2. Open Field Test (OFT)

OFT test was used to assess anxiety behaviors. Animals were allowed to freely explore in an open field box for 30 min after being placed in the middle zone of the arena, with the experimenter out of the animal’s sight. After each run, any feces were removed, and the arena was thoroughly cleaned. Analysis of the recording was completed by measurement of time spent and distance traveled by each mouse in the periphery and the center (the most anxiogenic area) of the arena.

### 2.3. Total RNA Isolation and Real-Time Quantitative PCR (RT-qPCR)

Total RNAs were extracted from the OB and rostral FC using TRI Reagent^®®^ (TR 118) (Molecular Research Center, Inc., Cincinnati, OH, USA), and cDNAs were synthesized with a RevertAid First Strand cDNA Synthesis Kit (Thermo Scientific, Waltham, MA, USA). Gene primers (Appendix A) designed in Primer3 with BLAST sequence verification (TAG Copenhagen AS, Frederiksberg, Denmark) were mixed with 5x HOT FIREPol^®®^ EvaGreen^®®^ qPCR Supermix (Solis BioDyne, Tartu, Estonia) and RT-qPCR was performed on equipment with QuantStudio 12KFlex Software v.1.2.2 (Applied Biosystems, Waltham, MA, USA) according to the respective manufacturer’s instructions. Quantification was performed by first normalizing Ct values of target genes to the reference gene (*Gapdh*), and mRNA levels were expressed as exponential fold-changes against the smallest ΔCt value (2^−ΔΔCt^).

### 2.4. Immunohistochemistry (IHC)

Mice were euthanized with CO_2_. Floating sagittal sections of 4% paraformaldehyde-fixed brains (*n* = 3–5 per genotype) in 40 µm thickness were washed in PBS + 0.5% triton X-100 for 15 min, antibody against ionic calcium-binding adaptor molecule-1 (Iba1, #SKL6615, Wako) diluted at 1:1000 in blocking buffer (PBS with 10% goat serum + 1% BSA + 0.1% tween20 + 0.3M glycine) was added and incubated at 4 °C overnight. After antibody removal, the sections were washed for 15 min × 3 times with PBS. Goat anti-rabbit secondary antibody conjugated with Alexa488 (#155272, Jackson ImmunoResearch, West Grove, PA, USA) was diluted at 1:500 in PBS and incubated with the sections at room temperature for 2 h. Following PBS washing, the sections were mounted on glass slides with fluorescence-protecting mounting medium (Fluoromount, Thermo Fisher, Waltham, MA, USA). Images of 800 × 800 pixels in 0.5 × 40 µm Z stack, seven images per OB and FC region per genotype were taken by a laser scanning confocal microscope (Olympus FV1200MPE) with a CCD camera under 60× magnification.

### 2.5. Flow Cytometry

Dissected brain tissues were homogenized through 70 µm cell strainers (#352350, BD Bioscience) in ice-cold flow buffer (PBS with 1% fetal calf serum). Isolated cells were blocked with 10% rat serum in ice-cold PBS for 60 min and stained with 0.5 µL mAb flow markers (all Biolegend, San Diego, CA, USA) of anti-mouse MHCII-PE (#107608), CD11b-PerCP/Cy5.5 (#101228), CD45-PECy7 (#103114), and CD206-APC (#141708) as well as isotype control rat antibodies IgG2b-PE (#400607), IgG2b-PerCP/Cy5.5 (#400631), IgG2b-PECy7 (#400617), and IgG2a-APC (#400511) in a flow buffer for 1 h. Cells were then fixed with 4% paraformaldehyde followed by permeabilization with 0.05% triton X-100 in PBS at 4 °C. After washings, cells were incubated with an anti-vesicular glutamate transporter-2 (VGLUT2) mAb-Alexa488 (#MAB5504A4, Millipore, Burlington, MA, USA) or isotype ctrl mAb-Alexa488 (#400132, Biolegend) for 60 min at 4 °C. Washed and resuspended cells were acquired with a Fortessa flow cytometer (BD Bioscience). Data were analyzed using Kaluza v2.1 software (Beckman Coulter, Brea, CA, USA). Macrophages were defined as CD45highCD11bhigh and microglia as CD45lowCD11bhigh cells. Cell numbers of microglia and macrophages normalized against equal amounts of collected brain cells were quantified, and percentages were calculated.

### 2.6. Statistical Analysis

Data were analyzed using GraphPad Prism 8.1.2. Main effects and interactions (genotype × brain regions) were determined using a two-way analysis of variance (ANOVA) with Tukey’s test for post-hoc multiple comparisons. Mann–Whitney test was used to assess nonparametric data. Statistical significance was set at *p* < 0.05 with mean and standard error of mean (SEM) reported.

## 3. Results

### 3.1. Young Adult FAD Mice Showed Enhanced Anxiety

We first measured the level of anxiety among 2 mo 5 × FAD and littermate WT male mice in OFT and found that the total number of moves (Figure 1A) and distance traveled (Figure 1B) as measures of locomotor activity were significantly decreased in 5 × FAD mice compared to WT mice. The 5 × FAD mice further showed a decreased distance traveled (Figure 1C) and time spent (Figure 1D) in the center, indicating enhanced anxiety in the early stage of amyloid pathology.

### 3.2. Pro-Inflammatory Cytokines Were Upregulated in the OB of Young 5 × FAD Male Mice

We further compared region-dependent expressions of pro-inflammatory cytokines in 2 mo 5 × FAD mice and found that *Il1b* and *Tnf* mRNAs were both significantly upregulated in the OB of young 5 × FAD males compared to WTs (Figure 2A), while no changes in these cytokines were observed in the FC (Figure 2B). We did not observe significant changes in other cytokines such as *Il18* and *Il6* (data not shown). Besides, these cytokines were expressed at extremely higher levels in the OB than in the FC of the young cohort (Appendix A). These results support the idea that the OB is an important early neuroinflammatory indicator in AD.

### 3.3. Tnf Was Associated with Anxiety in Young 5 × FAD Male Mice

TNF-α plays a vital role in the progression of cognitive decline as well as anxiety [19]; targeting TNF-α signaling was effective in modulating anxiety in mice [20]. Thus, we evaluated correlations between the parameters of anxiety and the *Tnf* mRNA levels in both the OB and FC of WT and 5 × FAD young male mice. Interestingly, we found significant negative correlations of *Tnf* mRNA levels in the OB with distance traveled in the center (*r* = −0.8529, *p* = 0.0147) (Figure 3A) and total number of moves (*r* = −0.7707, *p* = 0.04263) (Figure 3C) in 5 × FAD mice but not WTs, while no significant relationships were observed in the FC of both genotypes (Figure 3B,D). These results suggest the involvement of OB-bound neuroinflammation in the development of anxiety in early AD.

### 3.4. Microglial Activation and Morphological Changes in the OB and FC of Young 5 × FAD Male Mice

We further explored region-dependent changes in microglial activation and morphology in 2 mo 5 × FAD male mice. First, we compared IBA-1 intensity between WT and 5 × FAD mice and noticed that it was significantly elevated in the OB of 5 × FAD mice compared to WTs (Figure 4A,B), while no difference was found in the FC (Figure 4C,D), indicating increased microglial activation or abundancy in the OB of 5 × FAD mice. Next, we measured the morphometrics of microglia in detail. Interestingly, we found that the ramification indices of microglia were increased in both the OB (Figure 4B) and FC (Figure 4D) of 5 × FAD mice compared to WTs, especially in the FC. Furthermore, the average microglial primary branch (cable) length was higher in the OB of 5 × FAD mice compared to WTs (Figure 4B), with no change in the FC (Figure 4D). Together, these data suggest that microglia were already primed in the early pathological stage of AD; furthermore, OB microglia were more activated, whereas their FC counterparts showed a milder priming response.

### 3.5. Elevated Pro-Inflammatory Cytokines in the FC of Older 5 × FAD Male Mice

Since we observed increased microglial ramification but not pro-inflammatory cytokine production in the FC of young 5 × FAD mice, suggesting non-inflammatory activation of microglia in the FC at an early stage, we assumed neuroinflammation in the FC may become more apparent at a later stage. Hence, we further checked the levels of pro-inflammatory cytokines in the FC of 5~6-mo male mice. Agreeably, *Il1b*, *Il18*, and *Tnf* mRNA levels were all significantly increased in the FC of these older 5 × FAD males compared to WTs (Figure 5). We did not have sufficient OB tissues from this cohort, however, since we used these OB tissues for flow cytometry to align with the previous IHC result and to further characterize OB microglial synaptic functions in more detail.

### 3.6. Microglial Presynaptic Pruning Was Enhanced in the OB of Older 5 × FAD Male Mice

We next quantified OB microglia and their synaptic pruning activity in the 5~6 mo cohort by flow cytometry (Figure 6). The step-wise gating strategy to discern whole OB cell populations, including myeloid cells (i.e., macrophages and microglia) and microglial MHCII+/CD206+ subpopulations, are shown in representative dot plots in Figure 6A. The staining of respective isotype controls is represented in Appendix A. Microglial and macrophagic numbers were calculated among the equal number of total brain cells. Notably, microglia and macrophages were more abundant in the OB compared to the FC in both genotypes (Figure 6B,C). We also studied the cerebellum (CBM) and found a lower cerebellar microglial number compared to the other two brain regions (Figure 6B). Moreover, MHCII+ microglia were also increased in the OB compared to the other regions (Figure 6D). However, no genotypic differences in microglial and macrophagic abundances were observed (Figure 6B,D). Nevertheless, when we further measured VGLUT2 engulfment to evaluate the presynaptic pruning activity of microglia, we found that VGLUT2 abundancy was significantly increased in the OB of 5 × FAD males compared to WTs, but not in the FC and CBM (Figure 6E). Besides, OB microglia had more enriched VGLUT2 abundancy compared to their FC counterparts of both genotypes (Figure 6E). There was also an interactive effect between genotype and region in the VGLUT2 synaptic pruning (*F* (1, 22) = 6.289, *p* = 0.0069).

## 4. Discussion

Mechanisms contributing to neuropsychiatric symptoms such as anxiety in early AD are not well understood. The current study explored region-dependent neuroinflammation and microglial synaptic pruning and morphological changes in the 5 × FAD mouse model. Our most interesting finding was the OB-specific neuroinflammation at early-stage 5 × FAD, associating anxiety endophenotype with olfactory microglial activation. Anxiety and olfaction may mutually affect each other, and the relevant brain regions regulating these processes are among the most vulnerable areas for amyloidal pathologies and neuroinflammation in AD [8]. Our work thus provides preliminary evidence on the potential linkage between anxiety and olfactory neuroinflammation in early AD.

It is worth mentioning that in the 5 × FAD model, amyloidal pathologies are known to occur earlier in female mice than in males [13,21,22]. Hence, most studies have been prioritized on female mice, while little is known about male mice so far, driving us to focus on male mice here. We first found that anxiety occurred already in 2 mo 5 × FAD mice, a stage before cognitive impairment occurs in this model. This is in line with multiple earlier reports on human patients, denoting anxiety as a risk for AD progression in prodromal patients and accelerated cognitive decline in diagnosed patients [3,4,5,6]. We next discovered that the OB had increased *Il1b* and *Tnf* expressions compared to the FC, and furthermore, these cytokines were more upregulated in the OB of young 5 × FAD males than in WTs, which was phenocopied in the FC of 5 × FAD males at the later 5~6 mo stage. We also found that an increase in pro-inflammatory cytokine *Tnf* mRNA level in the OB was associated with enhanced anxiety in 2 mo 5 × FAD males. These suggest that baseline inflammatory status is higher in the OB compared to the FC, and olfactory neuroinflammation may be a valuable predictive biomarker for AD progression. Corroborating our findings, a previous study showed that a decline in olfactory function occurred earlier than other pathological changes in 5 × FAD mice [13].

In AD, glia-produced inflammatory cytokines and chemokines cause cyclic gliosis, which in turn exacerbates amyloidosis and other AD neurodegenerative pathologies [23,24]. Microglia and brain-associated macrophages, as well as their effector molecules, are the most important myeloid mediators of neuroinflammation in the AD brain [25,26,27]. Microglia react to pro-inflammatory signals with morphological and functional changes [28]. We characterized microglial morphometrics and observed increased microglial IBA1 intensity and ramification in the OB of 2 mo FAD male mice compared to WTs, hinting at more prominent microglial priming and supporting the above-described enhanced neuroinflammation in the OB at this stage. Additionally, FC microglia were also more ramified in 2 mo 5 × FAD males.

Microglial activation was found to occur before plaque deposition in 5 × FAD mice [18]. Enhanced microglial IBA1 intensity observed in IHC and increased pro-inflammatory cytokines observed in QPCR can be a result of both elevated expressions of these molecules in primed microglia and increased microgliosis. To answer this, we quantitatively measured the numbers of microglia as well as brain macrophages by flow cytometry. Firstly, we observed that the OB had increased amounts of microglia (particularly MHCII+ subtype) and macrophages as compared to other brain regions, in line with the higher inflammatory cytokine status of the OB at an early stage. Secondly, as no differences in myeloid cell numbers between WTs and 5 × FAD mice were found, microgliosis was unlikely to appear yet at an early stage of 5 × FAD, and molecular synthesis was probably promoted in early-stage primed microglia.

Importantly, neuroinflammation affects other fundamental microglial neuro-modulatory functions, such as synaptic pruning, a key mechanism underlying microglia-induced cognitive deficit at a more developed stage of AD [25,26]. We hence checked VGLUT2 engulfment by microglia in the OB and other brain regions of 5~6 mo 5 × FAD mice. Notably, we found that this pruning activity was more enhanced in 5 × FAD mice compared to WTs; OB microglia were also more active compared to their FC and CBM counterparts in both genotypes, which is in line with the higher expression of cytokines and IBA1 in the OB. Several lines of research corroborate our current findings. OB microglia are known to be active in synaptic pruning in healthy adult mice [29,30,31]. Activation of microglia under chronic psychosocial stress also causes aberrant synaptic pruning and imbalanced excitatory/inhibitory neurotransmission in the limbic regions, resulting in enhanced anxiety and depressive-like behaviors in rodents [32]. Furthermore, a lot of research has consistently demonstrated that elevated synaptic pruning and remodeling by activated microglia and microglia-derived cytokines, such as IL-1β and TNF-α, play detrimental roles in cognition in both mouse models and patients of AD [25,27,33,34].

Thus, our findings overall hint at the importance of OB microglia in contribution to behavioral deficits in early AD. It is plausible that OB microglia-associated synaptic remodeling and neuroinflammation contribute to both affective and cognitive deficits in AD, which, surprisingly, has not been thoroughly studied so far. Therefore, it would be imperative to develop further investigation beyond our current limited one, addressing critical questions such as the age- and sex-dependent association of OB microglial or other glial activation with olfaction-related behavioral dysfunctions in AD mouse models.

## 5. Conclusions

In summary, our preliminary findings reveal enhanced anxiety along with olfactory microglial activation in the early stage of an AD mouse model, highlighting the more active synaptic pruning of olfactory microglia in this model. Although our evidence is limited, it provides a first hint on the OB-specific neuroinflammation as a sensitive biomarker in early-stage AD.

## Figures and Tables

**Figure 1 biology-11-00938-f001:**
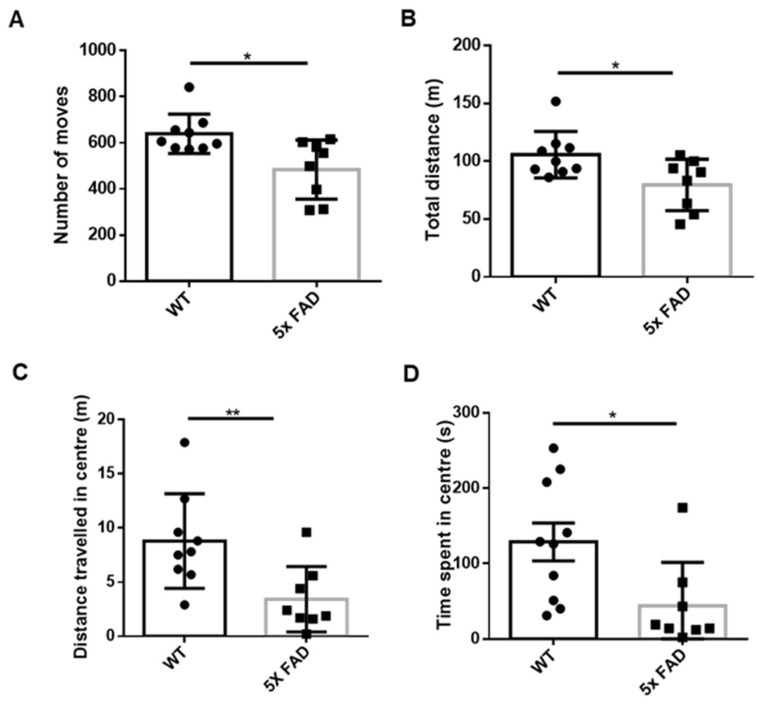
**Enhanced anxiety in 2 mo 5 × FAD mice at early neurodegenerative stage**. (**A**) Total number of moves; (**B**) total distance traveled; (**C**) distance traveled in the center; and (**D**) time spent in the center of an open field were measured. The 5 × FAD mice showed reduced stay in the central zone compared to WT littermates (*n* = 8–10). Data expressed as mean ± SEM; * *p* < 0.05, ** *p* < 0.01 (Mann–Whitney test).

**Figure 2 biology-11-00938-f002:**
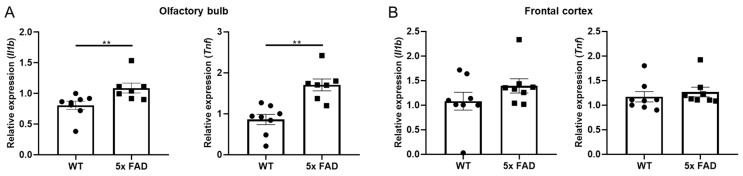
**Increased pro-inflammatory cytokines in the OB but not FC of 2 mo 5 × FAD male mice.***Il1b* and *Tnf* mRNA levels were measured by qPCR in the OB and FC (*n* = 7–8 per group). (**A**) *Il1b* and *Tnf* were more abundant in the OB of 5 × FAD mice (square dots) than WT littermates (circle dots); (**B**) *Il1b* and *Tnf* in the FC did not show genotypic differences. Data expressed as mean ± SEM; ** *p* < 0.01 (Mann-Whitney test).

**Figure 3 biology-11-00938-f003:**
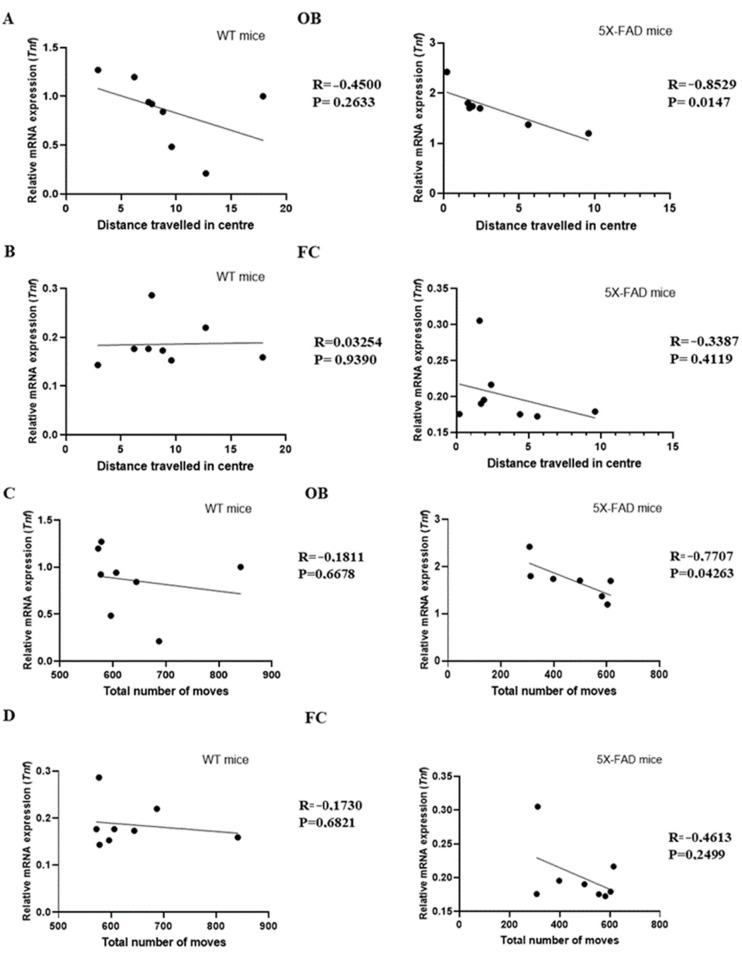
**Correlations of the pro-inflammatory cytokine *Tnf* in the OB with anxiety of 2 mo 5 × FAD male mice**. *Tnf* mRNA level in the OB but not FC was negatively correlated with distance traveled in the center (**A**,**B**) and total number of moves (**C**,**D**) in open field in 5 × FAD mice, while no correlations were present in WTs (*n* = 8) (Pearson’s test).

**Figure 4 biology-11-00938-f004:**
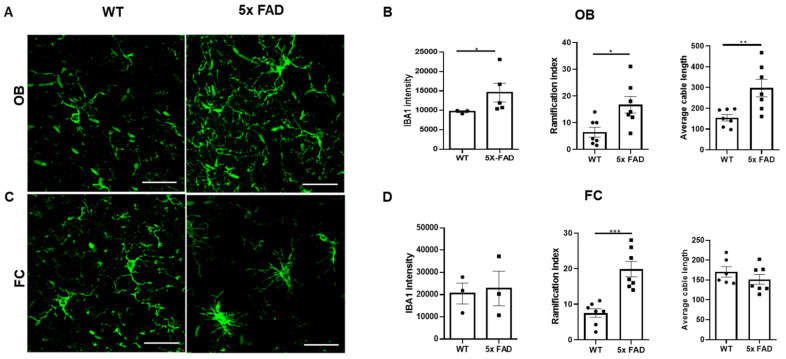
**Increased microglial intensity and ramification in the OB of 2 mo 5 × FAD male mice.** (**A**,**C**) Representative IHC images of Iba1 in the OB and FC. (**B**,**D**) Quantitative measurements of Iba1 showed increased microglial activation in the OB and enhanced ramification in the OB and the FC of 5 × FAD mice (square dots) compared to WTs (*n* = 3–7) (circle dots). Scale bar = 10 µm. Data expressed as mean ± SEM; * *p* < 0.05, ** *p* < 0.01, *** *p* < 0.001 (Mann-Whitney test).

**Figure 5 biology-11-00938-f005:**
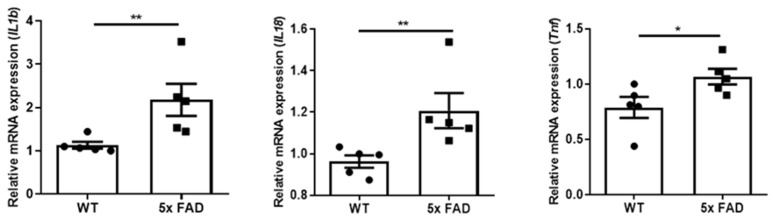
**Increased pro-inflammatory cytokines in the FC of 5~6 mo 5 × FAD male mice**. Increased mRNA levels of *Il1b*, *Il18*, and *Tn*f were observed in the FC of older 5 × FAD mice (square dots) compared to WTs (*n* = 5) (circle dots). Data expressed as mean ± SEM; * *p* < 0.05, ** *p* < 0.01 (Mann-Whitney test).

**Figure 6 biology-11-00938-f006:**
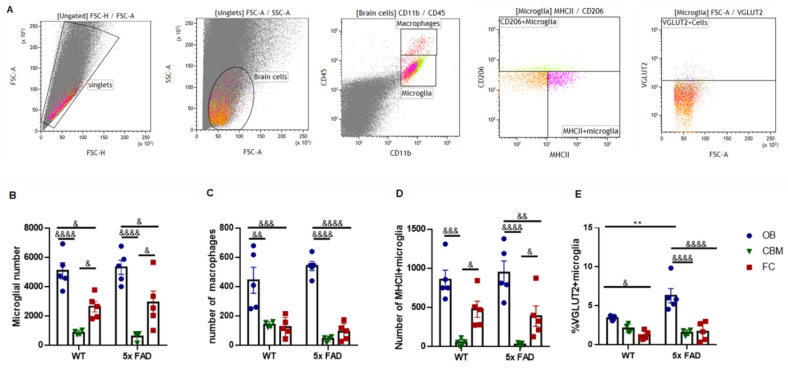
**OB microglia showed increased VGLUT2 phagocytosis in 5~6 mo FAD male mice.** (**A**) Representative graphs showing gating strategy for microglia, macrophages, and VGLUT2+ microglia. Numbers of microglia (**B**), macrophages (**C**), and MHCII+ microglia (**D**) among normalized total brain cells and percentage of VGLUT2+ microglia among total microglia (**E**) in different brain regions were quantitated. OB microglia were more abundant than other regional counterparts and showed enhanced VGLUT2 phagocytosis in 5 × FAD mice compared to WTs. * symbolizes significant genotypic difference while & regional difference. ** *p* < 0.01, & *p* < 0.05, && *p* < 0.01, &&& *p* < 0.001, &&&& *p* < 0.0001 (two-way ANOVA with Tukey’s test).

## Data Availability

Not applicable.

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
