# Peer review of "Enhanced Anxiety and Olfactory Microglial Activation in Early-Stage Familial Alzheimer’s Disease Mouse Model"

_biology, 2022, doi:10.3390/biology11060938_

Round 1

Reviewer 1 Report

This work studied the existence of enhanced anxiety and olfactory microglial activation in early-stage familial Alzheimer’s disease mouse model. There are some concerns in this manuscript as follows:

  • The meaning of the abbreviations should be clearly defined at their first mention (e.g. IL18, Tnf).
  • The abbreviation (Tnf) should be changed to (TNF-α) throughout the manuscript.
  • Try to avoid abbreviations (e.g. AD) in the key words.
  • The novel aspects of this study should be clarified in the “Introduction’ section.
  • The numbering of the subheadings should be revised.
  • It is advisable to perform histopathological examination of the brain sections to confirm the presence of the characteristic features of AD.
  • In figure 6A, the authors should provide images with a higher resolution.
  • I think that the conclusion was not sufficient. The possible clinical implications of the results of the present study should be clearly addressed.
  • The manuscript should be revised by English-naïve speaker to improve the quality of the language.

Author Response

Please see the attachment with graphs inside.

Reviewer 2 Report

The authors here report that the olfactory lobe is involved in early-stage Alzheimer’s disease in mouse

However, the manuscript raises many questions that remain unanswered.

  1. What is the source of TNF-alpha in the OB of the 5xFAD mouse ? Are TNF and other pro-inflammatory cytokines produced by microglia or is there an increased infiltration of CD45+ cells in the OB of 5XFAD mice? Specifically is there an increased infiltration of monocytes? Authors need to distinguish the resident macrophages from the circulating macrophages/monocytes.
  2. In Figure 2, authors should separate the OB and FC in to two separate graphs. In its current format, one can’t appreciate the difference in the cytokine expression, if any, between WT and 5xFAD mice at 2 months of age in the FC region.
  3. Have the authors tried to quantify the pro-inflammatory cytokines in the OB and FC of the mouse brain by ELISA? All claims are currently based on mRNA expression.
  4. Gating for microglia needs to be re done- Please refer to this paper (Zhou et al., 2017Microglia Polarization with M1/M2 Phenotype Changes in rd1 Mouse Model of Retinal Degeneration)I don’t understand how the authors have decided to bucket the single cells ?
  5. Is there an increase in microglia number in the OB of 5xFAD mice or is there more activation of the resident microglia? The authors need to distinguish between the two.
  6. Is there an age dependent association between anxiety and the brain region involved? The authors need to tease that apart.
  7. At 5-6 months, the authors report that they don’t have enough OB tissue to carry out mRNA analysis. I think this an important experiment to be done and the authors need to investigate this because their entire argument is based on the fact that Ob is involved in early stage anxiety and to rule out the involvement of OB in the later stages of the disease, one needs to do the experiment.

Author Response

The authors here report that the olfactory lobe is involved in early-stage Alzheimer’s disease in mouse

However, the manuscript raises many questions that remain unanswered.

Response: We are very grateful for your constructive comments and fully agree that our current preliminary work raises many important but unanswered questions. We have revised our manuscript according to your comments and humbly hope that we could increase the awareness of them through this current work. Below is our point-to-point reply to your comments.

1. What is the source of TNF-alpha in the OB of the 5xFAD mouse? Are TNF and other pro-inflammatory cytokines produced by microglia or is there an increased infiltration of CD45+ cells in the OB of 5XFAD mice? Specifically is there an increased infiltration of monocytes? Authors need to distinguish the resident macrophages from the circulating macrophages/monocytes.

Response:

We did not isolate microglia and other glia from 5xFAD mice to test by QPCR or ELISA, so we cannot directly answer whether microglia were the only resource of TNF-a and other pro-inflammatory cytokines in 5xFAD mice. In normal mice, Tnf and Il1b are mainly produced by microglia, as shown in the Stanford Brain RNAseq database: https://www.brainrnaseq.org/.

For infiltration of monocytes, the flow cytometric markers CD45 and CD11b that we used cannot differentiate macrophages from monocytes and we did not have monocyte-specific markers such as Ly6C. So, in Figure 6 where we measured macrophages, there may be or may not be monocytes inside. The total number of macrophages did not differ between WT and 5xFAD mice but we don’t know if there was increase of circulating monocytes in the OB of 5xFAD mice or not. In addition, when activated such as in AD condition, microglia and infiltrated macrophages are very similar in their phenotypes, making it very challenging to fully differentially distinguish them.    

2. In Figure 2, authors should separate the OB and FC in to two separate graphs. In its current format, one can’t appreciate the difference in the cytokine expression, if any, between WT and 5xFAD mice at 2 months of age in the FC region.

Response: Figure 2 results on the OB and FC are separated in the new version now. For regional differences in cytokine mRNA levels, we have moved them as Supplementary Figure 1 now.

3. Have the authors tried to quantify the pro-inflammatory cytokines in the OB and FC of the mouse brain by ELISA? All claims are currently based on mRNA expression.

Response: No, unfortunately we have only data on mRNA expression currently. This indeed educates us to check the protein levels in the future.  

4. Gating for microglia needs to be re done- Please refer to this paper (Zhou et al., 2017Microglia Polarization with M1/M2 Phenotype Changes in rd1 Mouse Model of Retinal Degeneration)I don’t understand how the authors have decided to bucket the single cells ?

 Response: For flow cytometric gating, we first used FS-H/FS-A dot plot to exclude cell aggregates and debris and then gated on single cells including microglia in FS/SS dot plot. Our previous gate on single cells in the FS-H/FS-A and FS/SS dot plots included all microglia already, but to prevent confusion by the reviewer that some cells might have been excluded, we have broadened the Singlet gate in new Figure 6A now. This however only includes more of other brain cell types (e.g., neurons and astrocytes) and does not affect microglial quantification. Gating on isotype staining is also provided in Supplementary figure 2 at the end of the manuscript. 

5. Is there an increase in microglia number in the OB of 5xFAD mice or is there more activation of the resident microglia? The authors need to distinguish between the two.

Response: Figure 6B shows that there was no increase in total microglial number in the OB of 5-m-o 5xFAD mice, but there was a relative increase (%) of MHCII+ microglia, indicating more activation.

6. Is there an age dependent association between anxiety and the brain region involved? The authors need to tease that apart.

Response: This is an excellent question. Based on our cytokine results in Figures 2 and 4, it seems that age did have an effect on cytokine production at least in the frontal cortex and this may in turn lead to age-dependent differences in behavior between 5xFAD and WT mice, which is indeed worth to pursue. Unfortunately, we did not have behavioral anxiety data on 5-m-o 5xFAD mice to compare the age-dependent association between anxiety and the brain regions of microglial activation. We have mentioned this limitation of our current work at the end of the Discussion in lines 352-358.

7. At 5-6 months, the authors report that they don’t have enough OB tissue to carry out mRNA analysis. I think this an important experiment to be done and the authors need to investigate this because their entire argument is based on the fact that Ob is involved in early stage anxiety and to rule out the involvement of OB in the later stages of the disease, one needs to do the experiment.

Response: Yes, we absolutely agree on this excellent point too. It is a current limitation of our work and we hope that we can more carefully study OB microglia in 5xFAD mice in age- and sex-dependent manners in the future.

Reviewer 3 Report

The article is interesting and worthy to be published. Howver, a small revision is needed:

1) The Authors should present their ideas about putative relationship between anxiety and worse odor discrimination, due to olfactory dysfunction, in early-stage AD patients.

2) Did the Authors observe worse odor discrimination in the 5xFAS population compared to WT mice ?

3) Do the parameters shown in Figure 1A-D provide evidence about increased anxiety in the 5xFAD population ?

4) The Authors should make more comments about the data shown in Figure 6A. Moreover, the Supplementary Figure 1 was not available to the Reviewer.

5) The Authors should present their ideas about research studies to be performed in the future.

Author Response

The article is interesting and worthy to be published. However, a small revision is needed:

Response: We sincerely thank the reviewer for this encouraging comment. Below is our point-to-point reply to your comments.

1) The Authors should present their ideas about putative relationship between anxiety and worse odor discrimination, due to olfactory dysfunction, in early-stage AD patients.

Response: We have rewritten the Introduction and Discussion, providing more elaboration of the relationship between anxiety and olfactory dysfunction that putatively may be disturbed in AD. Please see the lines 63-68 and 290-294.

2) Did the Authors observe worse odor discrimination in the 5xFAS population compared to WT mice?

Response: We have not tested odor discrimination and associated cognitive behavior in this preliminary study. In the literature, mixed findings on odor-related cognitive performances of 5xFAD mice are presented. While several studies found deficits in odor learning and memory in them but no odor detection impairments (PMID: 24596271, PMID: 28844596, PMID: 29017573), some other studies reported that olfactory detection and memory was intact in male or female mice (PMID: 26969629, PMID: 32522622).

3) Do the parameters shown in Figure 1A-D provide evidence about increased anxiety in the 5xFAD population?

Response: Yes, Figure 1C and 1D showed that 5xFAD mice had reduced travel distance and time spent in the central area of open field box, indicating enhanced anxiety as compared to WT littermates.

4) The Authors should make more comments about the data shown in Figure 6A. Moreover, the Supplementary Figure 1 was not available to the Reviewer.

Response: We apologize that Figure 6A was not fully explained and Supplementary Figure 1 missed to reach you. Figure 6A is described in more detail in lines 253-255 now. Supplementary Figure 1 is renamed as Supplementary Figure 2 now and we have added supplementary materials at the end of the manuscript to avoid missing them again. 

5) The Authors should present their ideas about research studies to be performed in the future.

Response: We have revised the end of the Discussion, pointing out our current limitation and suggesting more careful investigation to age- and sex-dependent association of OB microglial or other glial activation with olfaction-related behavioral dysfunctions in AD mouse models in the future. Please see the lines 352-358.

Round 2

Reviewer 2 Report

Some questions that I have raised have not been satisfactorily answered by the authors. Is it because the authors don't have access to the resources? 

Author Response

Dear reviewer,

For your questions number 1 and 3, indeed, we did not have enough OB and FC tissues to isolate microglia and other brain cells to measure cytokines by qPCR or ELISA in cell-specific manner with the last cohorts of mice. Regarding questions number 6 and 7, the coauthor Dr. Miriam Hickey is the owner of 5xFAD mice and she currently doesn’t have a suitable cohort of 2-mo and 6-mo male mice for us to repeat the age-paired comparisons on behaviors and the following brain tissue analyses. It will take us a long time to breed a new cohort to fulfill this requirement under the 3R principle, so we would like to report on current findings and make a more thorough investigation on this project in the future. We would like to sincerely apply for your understanding.

Best regards,

Li Tian